# Antarctic climate and ice sheet configuration during the Early Pliocene interglacial at 4.23 Ma

Nicholas R. Golledge[1,2], Zoë A. Thomas[3], Richard H. Levy[2], Edward G. W. Gasson[4],
Timothy R. Naish[1], Robert M. McKay[1], Douglas E. Kowalewski[5], and Christopher J. Fogwill[3]

[1]Antarctic Research Centre, Victoria University of Wellington, Wellington 6140, NZ
[2]GNS Science, Avalon, Lower Hutt 5011, New Zealand
[3]Climate Change Research Centre and PANGEA Research Centre, University of New South Wales, Sydney NSW 2052, Australia
[4]Department of Geography, The University of Sheffield, Sheffield S10 2TN, United Kingdom
[5]Department of Earth, Environment, and Physics, Worcester State University, Worcester, MA 01602, USA

*Correspondence to:* Nicholas Golledge (nicholas.golledge@vuw.ac.nz)

**Abstract.** The geometry of Antarctic ice sheets during warm periods of the geological past is difficult to determine from geological evidence, but is important to know because such reconstructions enable a more complete understanding of how the ice-sheet system responds to changes in climate. Here we investigate how Antarctica evolved under orbital and greenhouse gas conditions representative of an interglacial in the early Pliocene at 4.23 Ma, when southern hemisphere insolation reached a maximum. Using offline-coupled climate and ice-sheet models, together with a new synthesis of high-latitude palaeoenvironmental proxy data to define a likely climate envelope, we simulate a range of ice-sheet geometries and calculate their likely contribution to sea level. In addition, we use these simulations to investigate the processes by which the West and East Antarctic ice sheets respond to environmental forcings and the timescales over which these behaviours manifest. We conclude that the Antarctic ice sheet contributed 8.6±2.8 m to global sea level at this time, under an atmospheric $CO_2$ concentration identical to present (400 ppm). Warmer-than-present ocean temperatures led to the collapse of West Antarctica over centuries, whereas higher air temperatures initiated surface melting in parts of East Antarctica that over one to two millennia led to lowering of the ice-sheet surface, flotation of grounded margins in some areas, and retreat of the ice sheet into the Wilkes Subglacial Basin. The results show that regional variations in climate, ice-sheet geometry, and topography produce long-term sea-level contributions that are non-linear with respect to the applied forcings, and which under certain conditions exhibit threshold behaviour associated with behavioural tipping points.

## 1 Introduction

The response of the Antarctic ice sheets (AIS) to predicted future oceanic and atmospheric warming will dictate the magnitude of global sea level changes for millennia, yet the sensitivity of the AIS system is unclear, leading to a wide range of sea-level predictions for coming centuries (Ritz et al., 2015; Golledge et al., 2015; DeConto and Pollard, 2016). Simulations of future scenarios such as these are most credible when constrained by observations of past changes. Here we simulate the AIS during a warmer period of the geological past for which geological and paleoenvironmental data exist (Cook et al., 2013;

Miller et al., 2012; McKay et al., 2012) with which to verify simulations. Warm periods of the Pliocene (2.58 – 5.33 Ma) are considered some of the most appropriate analogues for future environmental conditions (Haywood et al., 2016), especially the early to mid Pliocene (4 – 5 Ma) when atmospheric $CO_2$ concentrations were in the range 365 – 415 ppm (Pagani et al., 2010), similar to present-day. Globally-averaged surface temperatures during Pliocene interglacials were 2 to >3°C warmer than present (Haywood et al., 2013), comparable to warmings anticipated by 2100 under mid-range emissions scenarios of the Intergovernmental Panel on Climate Change Fifth Assessment Report (Collins et al., 2013). Far-field sea-level records imply a collapse of the West Antarctic Ice Sheet (WAIS) together with partial loss of marine-based ice from the East Antarctic Ice Sheet (EAIS) at this time (Miller et al., 2012; Rovere et al., 2014). Geological records from close to the East Antarctic coast suggest that a portion of this sea-level contribution may have originated from the Wilkes Subglacial Basin (WSB), when the ice sheet margin is thought to have migrated hundreds of kilometers inland (Cook et al., 2013; Patterson et al., 2014).

Given the large scale of these inferred ice sheet reconfigurations during warm periods of the Pliocene, it may be inferred that Antarctica's marine-based ice sheets may be vulnerable to thresholds beyond which an abrupt change in state occurs (c.f. Levy et al., 2016; Aitken et al., 2016). If such thresholds exist, and were crossed during Pliocene glacial–interglacial transitions, simulations of ice-sheet evolution during this period may be able to provide insights into processes that could be relevant under future warmer climate scenarios. To investigate the sensitivity of Antarctica to climate–ice-sheet thresholds under warmer climates, we first run global and regional climate models to simulate changes in Antarctic oceanic and atmospheric temperatures under an orbital configuration representative of an interglacial in the early Pliocene at 4.23 Ma, when austral insolation reached a maximum, and atmospheric $CO_2$ concentration was around 400 ppm (Figs. 1 & 2). We focus on this period, rather than the commonly investigated 'PRISM' interval of the mid-Pliocene (Haywood et al., 2013), as the higher insolation and atmospheric $CO_2$ concentration at this earlier time may have resulted in a warmer climate and a smaller ice sheet than the later period, which may help reconcile far-field sea-level records for the period. Outputs from these climate simulations are then used as inputs to ice-sheet model experiments that use the Parallel Ice Sheet Model, a fully-coupled ice-sheet/ice-shelf model (Bueler and Brown, 2009; Winkelmann et al., 2010). These simulations allow us to quantify the continental-scale changes that take place under the prescribed climatology, and the consequent contribution to sea level from the AIS. Although the peak climatic conditions we simulate with the RCM most likely only persisted for 1–2 kyr, we run our simulations for 10 kyr in order to quantify how the pertubed ice sheet might evolve if such warm conditions were maintained. Although this does not realistically reflect climatic forcings associated with orbital cyclicities, and implies that our sea-level estimates may be upper bounds for the prescribed climates, it is nonetheless a useful approach for the study of 'unforced' ice sheet oscillations, or 'tipping points' (Thomas, 2016). In the final part of our study, therefore, we use our long-term ice-sheet simulations to investigate whether evidence for tipping points can be seen, and if so, the mechanisms that give rise to them.

## 2  Methods

### 2.1  Climate and ocean inputs.

We use spatially-distributed air temperature, precipitation, and sea-surface temperatures from an established regional scale climate model (RCM) coupled with the GENESIS version 3.0 Global Climate Model (Thompson and Pollard, 1997; Pal et al., 2007). The GCM parameterises fluxes between the land surface boundary and the free atmosphere and includes detailed representations of snow and land ice. Outputs from the GENESIS model have been validated against observed polar climates and ice-sheet mass balance (Thompson and Pollard, 1997), and are therefore considered reliable for studies such as ours. Our regional climate model is RegCM3, which takes time-dependent lateral boundary conditions from the GCM. It simulates atmospheric dynamics, radiative transfer and precipitation, and includes a representation of the open ocean (Pal et al., 2007). RCM simulations at 40 km resolution are initialised from coarser resolution (T31) GCM outputs. We simulate the climate during a peak-insolation interglacial at 4.23 Ma based on an orbital configuration representative of that period (Fig. 1), together with a greenhouse gas concentration of 400 ppm. Although previous work has suggested that the relative role of insolation and $CO_2$ in driving warmer climates varies through time, and that high insolation in one hemisphere may not necessary lead to elevated temperatures at the same latitudes (Yin and Berger, 2012), palaeo-environmental proxies for this period are consistent with this being an interval of peak warmth (see 'Paleoenvironmental proxies' below) and fundamentally the ice-sheet responds only to the integrated warming anomaly, not insolation or radiative forcing in isolation.

Temperature and precipitation fields from the Pliocene RCM simulation are used as inputs to the ice-sheet model by calculating the anomalies from a present-day RCM simulation and then adding these to the present-day temperature and precipitation fields used in model initialisation and spinup (Figs. 2–4). This approach is preferable to one that uses the RCM fields directly, since the RCM does not simulate the present-day state precisely, when compared to obervational data (Fig. 2). Since the climate model also uses an ice-sheet topography in which WAIS is already removed, we use a standard lapse rate of 8 K km$^{-1}$ to adjust the simulated temperatures based on the elevation difference from our present-day ice geometry. Despite this adjustment, however, it is possible that the air temperatures in these areas may be anomalously warm as a result of ocean–atmosphere heat exchange that would not occur if the removed ice were present in the RCM simulation. This may have an influence of the retreat rate of West Antarctica in our simulations, but since our focus lies mainly on behaviour of the East Antarctic ice sheet we do not expect this issue to affect our findings. We do not apply any lapse rate adjustment to precipitation values on the basis that precipitation rates are likely to be controlled largely by synoptic patterns (which are captured by the RCM) rather than by a simple elevation-dependent cooling of the near-surface atmosphere. Since the RCM uses an ice-sheet geometry in which WAIS is absent it gives us a sea-surface temperature field for areas of our domain where grounded ice currently persists. However, these ocean temperatures are only used to calculate basal melt beneath ice shelves, not at the bed of grounded ice, and so do not affect the ice sheet unless the modelled grounding line retreats. For areas of grounded ice where we do not have RCM-derived ocean temperatures we prescribe a uniform value of 271.2 K, essentially the sea-water freezing point. This avoids potential errors that could be introduced by interpolating ocean fields landward, but we recognise that using a single, low, value rather than an interpolation may lead to underestimated basal melt in subglacial basins during ice sheet retreat.

Our offline, one-way climate–ice sheet model coupling is less satisfactory than a two-way coupling, but is computationally easier and allows first-order ice-sheet responses to climate perturbations to be investigated. However, we acknowledge that a fully-coupled set up might give more accurate ice-sheet and climate simulations.

## 2.2 Paleoenvironmental proxies

Local records of palaeo-temperature around the Antarctic continent are relatively sparse, yet those that exist, and can be chronologically-constrained with some certainty, offer a means by which climate model simulations can be evaluated. We compiled marine paleoenvironmental data from several circum-Antarctic locations that include drill cores from the western Ross Sea (CIROS-2, DVDP-10, AND-2A), Prydz Bay (ODP 1165), Kerguelen Plateau (ODP 751A), Wilkes Land (IODP U1361), and north western Antarctic Peninsula (ODP 1096) and geological outcrop at the Prydz Bay coast (Fig. 3). Data were

obtained from stratigraphic intervals that span 4.5 to 4.0 Ma (C3n.1r to lowermost C2Ar) which is characterised by relatively depleted values ($< 3.2‰$) in the benthic $\delta^{18}$O stack (Lisiecki and Raymo, 2005) (i.e. relatively warm glacials and interglacials; MIS CN1 and CN5 are two of the warmest interglacials in the early Pliocene). Sea water temperature estimates through this interval are based on a range of proxies including biological analogues and geochemical data that are described in detail below and which are summarised in Table 1.

Surface water temperatures from biological proxies are derived from diatom, silicoflagellate, and calcareous nannofossil data. Diatom assemblages within the target 'warm' interval at all of the drill sites contain, and are often dominated by, open ocean species *Thalassionema nitzschioides* and *Shionodiscus tetraoestrupii* (Whitehead and Bohaty, 2003a; Winter et al., 2010). Modern descendants of *T. nitzschiodes* and *S. tetraoestrupii* are restricted to areas north of the Polar Front where surface water temperatures are greater than 3.5 and 5.5°C respectively (Winter et al., 2010; Crosta et al., 2005). Numbers of *Azpeitia*

*spp*. also increase within the early Pliocene section at IODP Site U1361 (Cook et al., 2013) and reach a maximum abundance of 13%. Today *Azpeitia tabularis* is considered cold-tolerant in Southern Ocean waters but species abundance increases as water temperatures increase and generally exceeds 10% near the modern subtropical front (Romero et al., 2005), which suggests that early Pliocene summer sea surface temperatures at Site U1361 were as high as 10°C. Importantly, sea-ice associated diatom taxa are either rare to absent at most of the drill sites (Winter et al., 2010) or indicate much reduced sea ice coverage (Whitehead

et al., 2005). Overall, the diatom assemblages indicate that surface water temperatures at locations south of 60° in the early Pliocene were similar to those in modern subantarctic regions and that the circum-Antarctic ocean was sea-ice free throughout most or all of the annual cycle.

Relatively warm early Pliocene surface water temperatures implied by the diatom assemblages are supported by silicoflagellate data. In the modern marine environment the silicoflagellate genus *Distephanus* is dominant south of the Antarctic polar

front and the genus *Dictyocha* is rare or absent. The ratio of these genera in ancient sediment samples can be used to infer past ocean temperature. Dictyocha abundance at Ocean Drilling Programme Site 1165 increases between 4.3 and 4.4 Ma which suggests that mean annual SST increased to approximately 4°C during this interval (Whitehead and Bohaty, 2003b). A correlative interval in ODP Sites 748 and 751 contains a similar warming signal (Bohaty and Harwood, 1998).

Although temperature reconstructions based on geochemical proxies are rare, TEX$^L_{86}$ derived SST estimates of $5 \pm 4$°C were obtained from diatomite that was deposited between 4.5 and 4.2 Ma at the AND-1B drill site (McKay et al., 2012). These data support the temperature reconstructions based on biological proxies and indicate that summer surface water temperatures in the western Ross Sea were 5 to 6°C warmer than today. Furthermore, sedimentary facies indicate increased sediment input during interglacials (Motif 2b of McKay et al., 2009; Naish et al., 2009). This is interpreted as representing a warmer glacial regime (compared to the Late Pliocene) with more sediment-laden meltwater emanating from the margins of the EAIS and discharging into the Ross Sea.

Other evidence for warmth in the early Pliocene comes from fossil invertebrates and a unique assemblage of vertebrates preserved in the Sørsdal Formation, which crops out at Marine Plain in the Vestfold Hills (Fig. 3). The Formation comprises up to 7.2 metres of friable diatomaceous siltstone and sandstone with dark limestone lenses (Quilty et al., 2000). The age of the deposit is not well constrained, but diatom data suggest it was deposited between 4.4 Ma and 2.74 Ma, and most likely at the earlier end of this range, possible 4.2 to 4.1 Ma (Clark et al., 2013). Cetacean fossils include a species of dolphin, a beaked whale and baleen whale (Quilty et al., 2000; Fordyce et al., 2002). These noncryophilic species indicate that waters were ice free during summer (Fordyce et al., 2002). The Sørsdal Formation also contains the scallop *Austrochlamys anderssoni*. This and similar species of thick-shelled costate scallop thrived in Antarctic waters during interglacial episodes throughout the Neogene but disappeared during the Late Pliocene (Jonkers, 1998). Modern descendants of these thick-walled species do not live further south than sub-Antarctic regions where surface water temperatures are warmer than 5.5°C. Diatoms in the formation also indicate an annual range in SST from -1.8 to 5.0°C (Ref. (Whitehead et al., 2001)) and the absence of coccoliths indicates that summer SSTs were likely no higher than 5°C (Ref. (Whitehead et al., 2001)).

In summary, the evidence listed above indicates that summertime surface water temperatures in the early Pliocene were at least as high as 4-5°C during the warmest interglacial periods, and were likely warmer at 64°S (IODP site U1361). Evidence for minimal sea ice suggests that while winter water temperatures may have approached -1.8°C, summers were sea-ice free. Based on these data we infer that early Pliocene surface water temperatures in the oceans adjacent to Antarctica's continental ice sheets must have been higher than -1.8°C in the winter and at least 5°C in the summer; a mean annual temperature that was approximately 3°C warmer than today. Table 1 compares these proxy-based interpretations with values simulated by our climate model experiments.

## 2.3 The ice sheet model.

Our ice-sheet simulations use the Parallel Ice Sheet Model (PISM) version 0.6.3, an open-source, three-dimensional, thermo-dynamic, coupled ice-sheet/ice-shelf model. Both the model and our implementation of it are described in detail elsewhere (Bueler and Brown, 2009; Golledge et al., 2015; Aitken et al., 2016), so only a summary is given here. In brief, the model combines equations of the shallow-ice and shallow-shelf approximations (SIA and SSA respectively) for grounded ice, and uses the SSA for floating ice (Bueler and Brown, 2009). Superposing the SIA and SSA velocity solutions allows basal sliding to be simulated according to the 'dragging shelf' approach (Bueler and Brown, 2009), and enables a consistent treatment of stress regime across the grounded to floating ice transition (Winkelmann et al., 2010). Ice streams develop naturally as a con-

sequence of plastic failure of saturated basal till (Schoof, 2006), depending on the thermal regime and volume of water at the ice sheet bed. Because we employ a large number of simulations in our study, we adopt a relatively coarse model grid of 20 km. Mesh-dependence of results can be an important issue under certain circumstances (Martin et al., 2015), but in previous work we have shown that this is not the case in our simulations (Golledge et al., 2015). Furthermore, the principal aim of our

experiments is to identify differences between scenarios, rather than to define absolute ice geometries, and for this purpose we believe our approach is both suitable and robust.

Grounding-line migration is a key component of Antarctic ice sheet simulations, and is a much-debated modelling challenge. By default we adopt here a novel grounding-line scheme that uses a sub-grid interpolation method to more accurately track migrations (Feldmann et al., 2014). In this scheme we allow the sub-grid interpolation of driving stress at the bed, and the

calculation of one-sided derivatives to better characterise the ice-sheet / shelf junction. We run duplicate experiments both with and without an additional interpolation scheme that allows basal melt rates to be smoothly propagated across the grounding line from the first floating cell upglacier to the last grounded cell (Feldmann et al., 2014; Feldmann and Levermann, 2015). This mechanism tends to accelerate grounding-line retreat, leading to rapid mass loss in the initial centuries of the experiments, and greater long-term (near-equilibrium) sea-level-equivalent mass loss from the simulated ice sheets (Table 2). It is conceptu-

ally supported by geophysical studies of modern-day grounding lines that infer an 'estuarine'-type environment at ice-stream grounding zones (Horgan et al., 2013), and although less catastrophic than the cliff-collapse mechanism used in other models (Pollard et al., 2015; DeConto and Pollard, 2016) the approach is consistent with recent grounding-line process analyses (Gladstone et al., 2016). Basal melt beneath ice shelves (and at the grounding line) is calculated from a three-equation model that uses temperature, salinity, and pressure to determine the freezing point in the boundary layer (Hellmer et al., 1998; Holland

and Jenkins, 1999). Depending on whether there is melt, freeze-on, or neither, a different approximation of the temperature at the ice shelf base is used. Outputs from RegCM3 do not include sub-surface ocean temperatures, and furthermore, PISM is currently set up to read in just a single ocean temperature field. Consequently we are not able to investigate how vertical differences in ocean warming may impact the ice sheet. Our model does, however, allow the pressure effects of deeper bathymetry to be accounted for, giving rise to enhanced basal melt near grounding lines and less melt in central and outer sectors of the

shelf (cf. Golledge et al., 2017, Fig. S5).

Surface mass balance depends on monthly climatological data and a positive degree-day model that tracks snow thickness and allows for melting of snow and ice at 3 and 8 mm / °C / day respectively. We incorporate a white noise signal (normally-distributed, mean zero random temperature increment) into the calculation of daily temperature variations. The standard deviation of daily temperature variability is set at 2°C, somewhat lower than the commonly employed value of 5°C, on

the basis that the later has a tendency to overestimate melt (Seguinot, 2013; Rogozhina and Rau, 2014). Surface temperatures are adjusted for elevation according to an altitudinal lapse rate of -8 K / km, and a refreezing coefficient of 0.6 is used to mimic meltwater capture within the snowpack.

## 2.4   Experimental methods.

There are two approaches to dealing with ice model parameter uncertainty in the kind of study we present here. One approach undertakes thousands of low-resolution experiments (a large ensemble) with incremental changes in each of several key parameters, such as flow enhancement factors. The results are then subsequently analysed with respect to observational constraints to establish which ensemble members are consistent with the data. We do not adopt this kind of approach. Instead we follow a more targeted methodology in which model parameter choice is incrementally refined through an iterative procedure in which we constrain our model to fit the present-day ice sheet geometry and surface velocity field. To achieve a good fit we adjust ice flow parameters based on expert judgement, not in an unguided manner as is done with ensemble approaches. We start with an initial 20 yr smoothing run that uses only the shallow-ice approximation to derive velocities. This removes spurious surface irregularities. Secondly we implement a 150,000 yr run in which the ice sheet geometry is held fixed but where internal thermal fields are allowed to evolve. The third phase uses a 25,000 yr simulation in which full model physics are employed and the ice sheet is allowed to evolve on all boundaries, that is, it is entirely unconstrained. Through carefully guided parameter iteration our procedure results is a spun-up, thermally and dynamically equilibrated ice sheet simulation that is the best fit to observational constraints that is possible by tuning available model parameters (Golledge et al., 2015; Aitken et al., 2016). Thus although parameter uncertainty can be a large source of error under certain circumstances (cf. Yan et al., 2016), we argue that our approach significantly reduces this uncertainty prior to our undertaking the prognostic experimentation. All experiments start from the same spun-up present-day ice-sheet simulation.

We make an assumption regarding the state of the AIS prior to the interglacial at 4.23 Ma based on geological evidence that indicates ice extent greater than present during early Pliocene glacial episodes (Naish et al., 2009), implying that the AIS retreated through a present-day configuration during interglacial transitions. However, since the initial condition of the ice sheet is not known, our assumption of a present-day configuration may be erroneous. However, differences in initial geometry would most likely affect the rate at which new equilibria to the imposed climatologies occurred, rather than the geometries of those steady states. Consequently we consider our equilibrium simulations to be representative of the likely long-term Antarctic response to the prescribed orbital and greenhouse gas configuration. We use outputs from the RCM (described above) to define anomalies to our present-day climate grids (Comiso, 2000; Lenaerts et al., 2012). Simulations are run for 10 kyr, which is long enough for most simulations to approach a steady state (see 'Results' below). We first model ice sheet evolution using climate fields simulated by the RCM, using both implementations of the sub-grid grounding-line scheme described above. Then we explore the threshold response of the EAIS by running additional experiments with uniform increments of 1 and 2° C added to the air and sea-surface temperature fields, which address both a known cold bias in the RCM as well as mismatches between RCM and proxy-based temperature reconstructions. By bracketing a range of temperatures we are also able to investigate more easily the existence and sensitivity of the AIS to climate–ice sheet thresholds. To isolate the effects of the imposed climatic perturbations most clearly we keep sea level at its modern level, rather than the +10 to +30 m thought likely for the Pliocene (Miller et al., 2012; Rovere et al., 2014; Winnick and Caves, 2015). We note, however, that in separate experiments not shown

here the response of the ice sheet appears to be unaffected by sea level changes of these magnitudes. We run duplicates of all experiments using the two variants of the sub-grid grounding-line scheme described above.

## 2.5 Tipping Point analysis.

The aim of this technique is to analyse timeseries data (in our case ice mass evolution) to find early warning signs of impending tipping points. Such events are characterized by a non-linear response to an underlying forcing, based on the phenomenon of 'critical slowing down', and relies on the gradual shallowing and widening of the state-space of the system (also known as the 'basin of attraction') (Scheffer et al., 2009, 2012). This 'slowing down' can be mathematically detected by looking at the pattern of fluctuations in the short-term trends of the data before the threshold is passed (Dakos et al., 2008). This change in shape of the state-space allows the system to travel further from its point of equilibrium (van Nes and Scheffer, 2007), whereupon the system takes increasingly longer to recover from small pertubations (Dakos et al., 2012), which can be detected as an increase in the lag-1 autocorrelation and variance of the time series (Ives, 1995). A change in skewness may also be interpreted as a precursor to tipping (Guttal and Jayaprakash, 2008).

This tipping point analysis technique has been applied to a range of climate and palaeoclimate data using both natural archives and model outputs (Dakos et al., 2008; Lenton et al., 2012; Thomas et al., 2015; Thomas, 2016; Turney et al., 2015). Importantly, the early warning signs, or precursors, are only expected in the presence of two or more quasi-stationary states, separated by an unstable equilibrium, and are not expected in the case of tipping induced by stochastic fluctuations only, so their identification tends to imply a real threshold change. The method (Dakos et al., 2008; Guttal and Jayaprakash, 2008; Dakos et al., 2012) only analyses data preceding the tipping point. These data are detrended using a Gaussian kernel smoothing filter over a suitable bandwidth, so that long term trends are removed without overfitting the data. The resulting residuals are then measured for autocorrelation at lag-1 and variance over sliding windows of two different sizes. The Kendall tau rank correlation coefficient (Kendall, 1948) is used to provide a quantitative measure of the trend; this metric varies between +1 and -1, where higher values indicate a greater concordance of pairs and a stronger increasing trend. Importantly, it is the presence of a parallel increasing trend in both autocorrelation and variance, rather than the absolute values of the indicators, that indicates critical slowing down (Ditlevsen and Johnsen, 2010).

For the purposes of this study we make an important distinction between a tipping point and a threshold, which are sometimes used synonymously. We define a tipping point as a transient feature, whereas a threshold is non-temporal. During the evolution of an ice sheet under a constant forcing it may be that a point is reached in which the trajectory of evolution changes, i.e., the system 'tips' into a new state of behaviour. By analysing the timeseries data from a constant forcing experiment, tipping point analysis is able to not only show where genuine system instabilities occur, but also to provide information on the timescale over which this instability evolves. This is what the final part of our study investigates. However, this point of change, under a steady forcing, is not the same as the identification of a single temperature at which the ice sheet may be stable or unstable in a given area and over a discrete period of time. Consequently it is not possible from our tipping point analysis to provide information on a threshold, as the two phenomenon are simply different entities. A detailed explanation of tipping points in

Earth systems can be found in Thomas (2016), whereas threshold temperatures for individual Antarctic ice sheet catchments have been quantified in Golledge et al. (2017).

## 3 Results

For the first of our experiments, we run our RCM to simulate the early Pliocene climate at 4.23 Ma. The model yields spatially-variable annual air and sea-surface temperature changes whose means are approximately 5 and 2°C above present respectively,
for presently ice-covered areas of Antarctica (Figs. 3 & 4). The advantage of using a known period of the past such as this, rather than a forward projection, is that we can use empirical data to evaluate our modelled climate fields and to constrain our ice-sheet simulations. Proxy data suggest that our simulated climate may be 1–2°C cooler than actually occurred (Table 1), thus we run duplicate simulations allowing for additional warming of 1 and 2°C in both the atmosphere and ocean. The sea-level equivalent ice volume loss from Antarctica in these simulations ranges from 4 to 14 m (Table 2; Fig. 5). Assuming a
5 to 7 m contribution from the Greenland ice sheet (Koenig et al., 2015) and 0.5 to 1 m from thermal expansion of the oceans (assuming 1 to 3°C ocean warming) (Rugenstein et al., 2016), all of our simulations are consistent with a Pliocene sea level highstand range of approximately 10 to > 20 m reconstructed from proxy records (Miller et al., 2012; Rovere et al., 2014; Winnick and Caves, 2015). However, by considering likely air–ocean temperature relationships (Rugenstein et al., 2016), as well as empirical records of an absent WAIS and a retreated EAIS at this time (Naish et al., 2009; Cook et al., 2013), we
are able to identify two scenarios as most plausible. Figure 6**a, b** illustrates the modelled Antarctic ice-sheet geometry that arises under constant forcing with the Pliocene climate, modified to incorporate a +2°C air temperature bias and a 0 or 1°C bias in the ocean (see 'Methods'). In both cases we reproduce the smaller-than-present ice geometry, as well as the pattern of basal ice velocities that would have controlled bedrock erosion and sediment transport, that are necessary to be consistent with geological interpretations (Naish et al., 2009; Cook et al., 2013).
In both of the scenarios shown in Figure 6, substantial grounding-line retreat is evident in the WSB, but the Aurora and Recovery basins are less affected. To better understand the timescales and rates involved in retreat in the WSB, we take timeslice ice and bed geometries along the WSB flowline from the two simulations considered representative of Pliocene conditions at 4.23 Ma (Fig. 6**a, b**). In both scenarios, we find that margin retreat under constant climate forcing is punctuated by periods of stability. Figure 7**a** illustrates a gradual surface lowering at the margin of the ice sheet that continues for 2000
years before flotation and grounding-line retreat ensue. At this point, rapid retreat takes place across the deepest part of the WSB, and grounded ice is replaced by a floating ice shelf. Peak retreat rates are sustained for two to three centuries during this period (Fig. 7**c**) before declining. Continued retreat across the inner portion of the WSB proceeds more slowly, but is similarly punctuated by alternating episodes of acceleration and relative stability that continue for 7 to 8 kyr (Fig. 7**c**). Under warmer oceanic conditions (Fig. 7**b**) retreat proceeds more quickly, but initial destabilization from the seaward pinning point
still requires thinning and lowering of the ice-sheet margin, which in this case takes place over the first 1000 years.

Figure 7 illustrates that thresholds clearly exist in areas like the Wilkes subglacial basin where the topography beneath the ice sheet allows for rapid retreat after prolonged thinning under a warmer climate destabilises the grounded margin. However,

in order to robustly assess whether genuine 'tipping points' exist in the Antarctic ice-sheet system it is necessary to use a statistical treatment of the timeseries data in which statistical signatures of instability are sought. To do this we first plot sea-level equivalent whole-continent ice volume trends (Fig. 8**a, h**), using air and ocean temperature bias-corrections as described above. Viewing ice volume evolution of these six simulations in semi-log space it is evident that the modelled ice sheets evolve through three distinct phases. First, in all experiments there is a period of rapid mass loss in which the marine-based sectors of the WAIS collapse. This phase takes one to three centuries, depending on the magnitude of the applied forcing. The second

phase is characterised by a slower, but sustained, sea-level contribution from the EAIS, arising from both dynamic adjustment to the loss of fringing ice shelves, and from surface lowering resulting from negative mass balance in coastal areas. It is during this phase that trajectories begin to diverge, depending on the air temperature forcing applied. Using the unadjusted RCM air temperatures, the EAIS stabilises and actually accumulates sufficient mass from the associated increase in precipitation to offset domain-integrated losses. In this scenario, the ice sheet effectively removes water from global oceans and leads to a slight

lowering of sea level (Fig. 8**a**, blue line). Perturbing the climate with an additional one or two degrees of atmospheric warming, however, means that surface melt is no longer completely offset by increased precipitation, and the sea-level contribution from the ice sheet continues to increase. The third and final phase of ice-sheet evolution in our simulations takes place once ice in major subglacial basins begins to retreat. Interestingly, in terms of ice-sheet thresholds the key tipping point identified in the Wilkes Subglacial Basin appears to be triggered in very similar ways by both atmospheric and oceanic forcing (Fig. 8),

suggesting that several climate scenarios may produce equilibrium ice-sheet geometries that are consistent with the empirical constraints.

Using a sliding window to carry out tipping point analyses of these data we can establish whether or not statistical signatures of instability are evident during any of these three phases. We analysed six ice sheet trajectories that encompass the range of environmental conditions considered most consistent with RCM simulations and empirical constraints, using air temperature

biases of 0, 1, and 2°C and ocean temperature biases of 0 and 1°C added to the RCM-simulated climatologies. In these six experiments we found evidence of critical slowing down in three. The loss of WAIS occurs too quickly in our simulations to determine whether tipping points for that ice sheet exist or not. However, in the coolest of the six scenarios, using the unadjusted RCM values, we found evidence for a tipping point that leads to accelerated growth of the East Antarctic ice sheet (Fig. 8**a,** blue line). The next warmest scenario yields a stable EAIS (Fig. 8**a,** yellow line) with no clear tipping point, and

the two warmest scenarios (Fig. 8**h,** yellow and red lines) lead to such rapid mass loss that no tipping points are detected. In the two intermediate scenarios (Fig. 8**a,** red line; **h,** blue line), however, two obvious inflections in mass loss are visible. Focusing on these two scenarios, we employ sliding analysis windows of two different widths, 250 and 1500 years, to see if precursor signals indicative of the build up to a tipping point can be seen in advance of the actual threshold. Figure 8**b, c, i, j** illustrate the trends in autocorrelation at lag-1 (AR1) and variance (Var) for the two scenarios, overlain on the corresponding

mass loss trajectories. In all cases, analyses using the 250-year window (black lines) reveal abruptly increasing AR1 and Var values (early warning signals of critical slowing down) immediately preceding the tipping point. A change in skewness is also observed (Figure 9), which suggests that the system may have reached a region of asymmetry in its basin of attraction. What is surprising, however, is that even the analyses using the 1500-year sliding window (Fig. 8**b, c, i, j**, grey lines) show evidence

of increasing trends (Kendall tau values shown in Figure 9), suggesting that the system may be showing signs of instability far in advance of the observed tipping point, and on a timeframe comparable to the gradual thinning observed in the Wilkes subglacial basin analysis described above (Fig. 7).

To determine whether the results are sensitive to parameter choices such as the length of the smoothing bandwidth or sliding window size, we ran repeats of the analysis with a range of 25 smoothing bandwidths (ranging from 5 to 15% of the time-series length) and 25 sliding window sizes (ranging from 25 to 50% of the time-series length). The results are visualized using contour plots of the Kendall tau values of these repeats; a more homogenous colour indicates increased robustness (Fig. 9). In order to test the significance of these results we created a surrogate dataset by randomising the original data over five thousand permutations based on the null hypothesis that the data are generated by a stationary Gaussian linear stochastic process. This method guarantees the same amplitude distribution as the original time series, but removes any ordered structure or linear correlation (Theiler et al., 1992) and makes no assumption about the statistical distribution of the data. The Kendall tau values for autocorrelation at lag-1 and variance were computed for each of the surrogate time series, and the probability of making a Type I statistical error (false positive) for the original data was computed by comparing to the probability distribution of the surrogate data (Table 3 and Figure 10).

To understand the glaciological changes taking place as these thresholds are crossed, we divided the analysis time periods into 'precursor' and 'response' episodes, and calculated mean thinning rates across the ice sheet (Fig. 8**d, e, k, l**). There are clear indicators of ice sheet thinning in the major subglacial basins of East Antarctica, but when velocity and thinning rate data are extracted from two of the most dynamic areas there appears to be no clear distinction between the two episodes – in one scenario thinning and acceleration seem to occur during both the precursor and the response phases, whereas in the other scenario dynamic thinning takes place only during the response phase (Fig. 8**f, g, m, n**). What is clear, however, is that wherever ice-sheet thinning is observed, it is associated with an acceleration of ice flow at the same location. In our examples the lag between peak thinning rates and maximum velocities ranges from 200 to 1200 years.

## 4   Discussion

Far-field records of Pliocene sea-level changes are sparse, and their interpretation is complicated by glacio-isostatic effects and dynamic topography (Raymo et al., 2011; Rovere et al., 2014). Various techniques used to define an envelope of the likely Pliocene sea-level highstand (Miller et al., 2012; Rovere et al., 2014; Winnick and Caves, 2015) imply a GMSL contribution from the EAIS, but the uncertainties involved are sufficiently great that the sea-level contribution from East Antarctica remains hard to quantify, even though some loss from this ice sheet seems likely (Haywood et al., 2016). Novel isotope-enabled climate and ice-sheet modelling has allowed the likely GMSL contribution from Antarctica to be constrained to 3–12 m during the warmest parts of the mid-Pliocene, with an absolute maximum of 13 m (Gasson et al., 2016), which requires at least some loss of ice from East Antarctica. The 4.23 Ma interglacial in our study was likely warmer than mid-Pliocene interglacials, however, suggesting mid-Pliocene sea-level estimates (Winnick and Caves, 2015; Gasson et al., 2016) should be treated as minima in our comparison. Warmer air temperatures during all Pliocene interglacials most likely led to increased precipitation in East

Antarctica, thickening the ice-sheet interior (Yamane et al., 2015). Consequently, any sea-level-equivalent mass loss from the EAIS must have been over and above these mass gains, perhaps supporting geologically-based inferences of large-scale ice-sheet margin retreat in areas such as the Wilkes Subglacial Basin (Cook et al., 2013).

Our study set out to shed further light on this period of uncertainty by investigating the sensitivity of the Antarctic ice-sheet to climate-related thresholds, with a particular focus on a peak-warmth interglacial of the early Pliocene period at 4.23 Ma. To establish the likely climatic conditions of this time we use global and regional climate models, but lean heavily on empirical
biogeochemical proxy data to validate these simulations. Because there are considerable uncertainties in both the modelled and proxy-inferred temperatures, we defined an envelope of air and ocean temperatures and ran a small ensemble of ice-sheet simulations to capture the range of possible responses. We find that, under the applied climate conditions, the Antarctic ice sheet contributed $8.6 \pm 2.8$ m to global mean sea level in the early Pliocene, consistent with some recent studies of the mid-Pliocene (Gasson et al., 2016; Yan et al., 2016) but higher than Pollard and DeConto (2009); de Boer et al. (2015) and lower
than Dolan et al. (2011); DeConto and Pollard (2016).

The reduction in AIS volume that we simulate arose from a loss of marine-based portions of WAIS, and an inland migration of the EAIS grounding line into the Wilkes subglacial basin. Lesser retreat occurred in the eastern Weddell Sea, in the Amery Basin, and at the margins of the Aurora subglacial basin. In our simulations it appears that the significant retreat in the WSB arises as a consequence of prolonged surface lowering due to atmospheric warming and increasing surface melt, which after
1000 to 2000 years allows flotation of the present-day ice margin and retreat from the primary pinning point that currently maintains stability of this sector. An increase in surface melting at this time is consistent with facies of this age in the Ross Sea that record increased deposition of terrigenous sediment Naish et al. (2009). Subsequent retreat proceeds rapidly into the inland-deepening basin, but is punctuated by periods of slower retreat where bedrock highs allow temporary stabilisation of the ice margin.
The delayed atmospheric warming control on ice-sheet behaviour that occurs in the WSB is also mirrored at the continental scale. Our whole-continent ice volume trajectories show that, over multi-millennial timescales, the magnitude of atmospheric warming is critical in determining whether or not ice-sheet mass loss exhibits tipping points (i.e., non-linear behaviour). We find the clearest evidence of tipping under climate scenarios that are moderate, rather than extreme. This is because of the sensitive interplay between the climate forcing that drives retreat and the topographic restraint afforded by pinning points that
aid stability. This balance is overwhelmed under extremely warm climates and ice-sheet retreat proceeds rapidly and linearly.

## 5   Conclusions

Recent studies have alluded to the idea that the Antarctic ice sheets may be susceptible to 'tipping points' in their stability that could give rise to non-linear behaviour and rapid contributions to sea level (Lenton, 2011; Mengel and Levermann, 2014). We have critically examined this by simulating the Antarctic climate and ice sheet system under warmer-than-present con-
ditions of a period of the past when geological evidence indicates a much reduced ice sheet, despite $CO_2$ levels similar to present. Our results indicate an Antarctic contribution to global sea level of approximately 8.5 m during the 4.23 Ma early

Pliocene interglacial, sourced primarily from West Antarctica and the WSB of East Antarctica. These findings are consistent with modern studies that implicate ocean thermal forcing as the dominant driver of marine-based ice-sheet retreat in West Antarctica (Joughin et al., 2010; Liu et al., 2015; Wouters et al., 2015), but we find in addition that over centennial to millennial timescales, atmospheric warming plays a key role in destabilizing sectors of East Antarctica (Golledge et al., 2015; DeConto and Pollard, 2016). Whereas the rate of ice-sheet retreat within EAIS subglacial basins is controlled by the interplay between ocean temperature, local bedrock elevation and ice thickness, initial destabilization appears to be governed by the prolonged and gradual surface lowering in response to warmer-than-present air temperatures. Under certain circumstances this long period of top-down melting constitutes the 'precursor' to the eventual tipping point, beyond which ice loss accelerates and produces a non-linear contribution to sea level. Whether such 'early warning' signals could be identified in the short timeseries of satellite-era records of the modern ice sheet remains an research area yet to be explored.

## 6  Code availability

The Parallel Ice Sheet Model is freely available as open-source code from the PISM github repository (git://github.com/pism/pism.git). RegCM3 is available from https://users.ictp.it/RegCNET/model.html. Bedrock topography and ice thickness data are from the BEDMAP2 compilation, available at http://www.antarctica.ac.uk//bas_research/our_research/az/bedmap2/. Information on surface mass balance data is available at http://www.projects.science.uu.nl/iceclimate/models/antarctica.php#racmo21. Air temperature and geothermal heat flux inputs were taken from the ALBMAP v1 compilation (Le Brocq et al., 2010) and can be downloaded from http://doi.pangaea.de/10.1594/PANGAEA.734145.

## 7  Data availability

The datasets generated and/or analysed during the current study are available from the corresponding author on reasonable request.

**Table 1.** Inferred high southern latitude winter, summer, and mean annual sea-surface temperatures from a range of proxy techniques compared to model predictions. Winter temperatures assume sea ice close to the coast but not in open water. Summer temperatures (and thus $T_{mean}$ also) are minima, and could be higher. Absolute sea-surface temperature predictions from the RCM shown for each site in italics. Although the considerable uncertainties associated with temperature inferences from palaeoecological proxies make precise comparisons impossible, the data generally indicate an underestimate of mean annual sea-surface temperatures in the RCM that averages approximately 1°C.

| Site | Lon (°E) | Lat (°N) | Winter (likely) | Summer (min.) | $T_{mean}$ (min.) | Reference |
|------|------|------|------|------|------|------|
| AND-1B | 167.083333 | -77.888889 | -1.8 | 5 | 1.6 | Winter et al. (2010); McKay et al. (2012) |
| | | | *-2.22* | *3.03* | *-0.22* | |
| CIROS-2 | 163.533333 | -77.683333 | -1.8 | 5 | 1.6 | Winter (1995); Levy et al. (2012) |
| | | | *-1.96* | *3.29* | *0.07* | |
| DVDP-10 | 163.511667 | -77.578612 | -1.8 | 5 | 1.6 | Winter (1995); Levy et al. (2012) |
| | | | *-1.96* | *3.29* | *0.08* | |
| ODP-1165 | 67.218733 | -64.379583 | 3 | 5 | 4 | Whitehead and Bohaty (2003b); Cook et al. (2014) |
| | | | *-0.02* | *7.19* | *3.54* | |
| IODP-U1361 | 143.886653 | -64.409547 | 3 | 5 | 4 | Cook et al. (2013) |
| | | | *-0.43* | *6.24* | *2.87* | |
| ODP-1096 | -76.96376 | -67.56681 | -1.8 | 5 | 1.6 | Winter and Iwai (2002) |
| | | | *-1.87* | *4.80* | *1.26* | |
| Vestfold Hills | 78.132 | -68.631111 | -1.8 | 5 | 1.6 | Whitehead et al. (2001) |
| | | | *-0.99* | *6.00* | *2.37* | |

**Table 2.** Mass loss in sea-level equivalent metres from the simulated Pliocene Antarctic ice sheet after 10 kyr, using environmental conditions from the regional climate model and augmented with a range of air and ocean temperature bias corrections. Numbers in parentheses represent duplicate experiments in which a more aggressive grounding-line scheme is employed (see 'Methods').

| | | sea-surface temperature bias | | |
|------|------|------|------|------|
| | | 0 | 1 | 2 |
| Air | 0 | 4 (4.6) | 6 (8.7) | 7.9 (12.2) |
| temp. | 1 | 5.6 (6.2) | 7.7 (9.4) | 8.8 (13) |
| bias | 2 | 7.7 (8.4) | 9.1 (10.9) | 10.3 (14.1) |

**Table 3.** Calculated significance statistics for two precursor metrics of tipping points. Kendall tau and $p$ values are shown for autocorrelation ('AR1') and variance ('Var') trends over 250 ('short') and 1500 ('long') sliding windows, for two scenarios in which distinct inflections occur in their mass loss timeseries. See Figure 10 for more information.

|  |  | Kendall Tau | $p$ value |
|---|---|---|---|
| Scenario 3 short | AR1 | 0.805 | <0.05 |
|  | Var | 0.626 | <0.12 |
| Scenario 3 long | AR1 | 0.588 | <0.15 |
|  | Var | 0.584 | <0.15 |
| Scenario 4 short | AR1 | 0.771 | <0.05 |
|  | Var | 0.911 | <0.01 |
| Scenario 4 long | AR1 | 0.656 | <0.1 |
|  | Var | 0.721 | <0.06 |

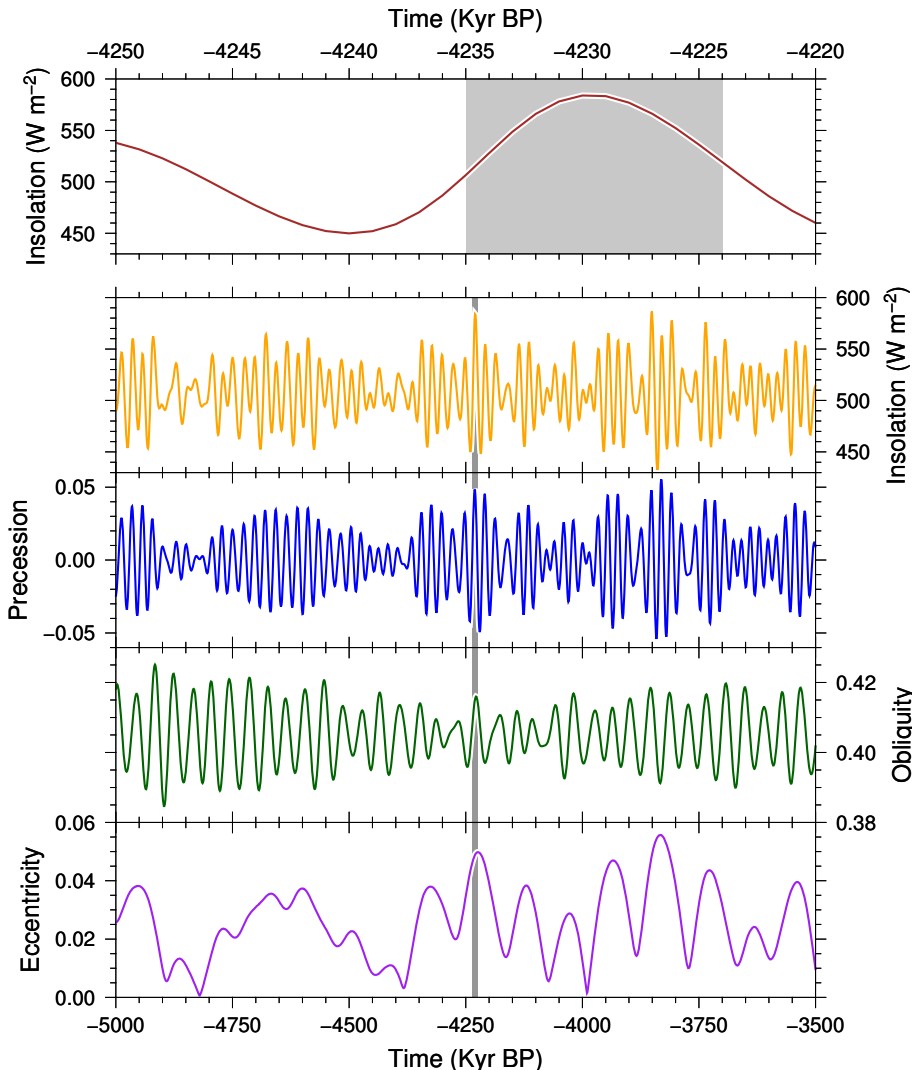

**Figure 1.** Orbital components and net January insolation at 80° S for the period 3.5 – 5 Ma. Top panel shows insolation for the interglacial in which peak Pliocene values occur (grey shading). The insolation peak occurs at 4.23 Ma. Data from Laskar et al. (2004).

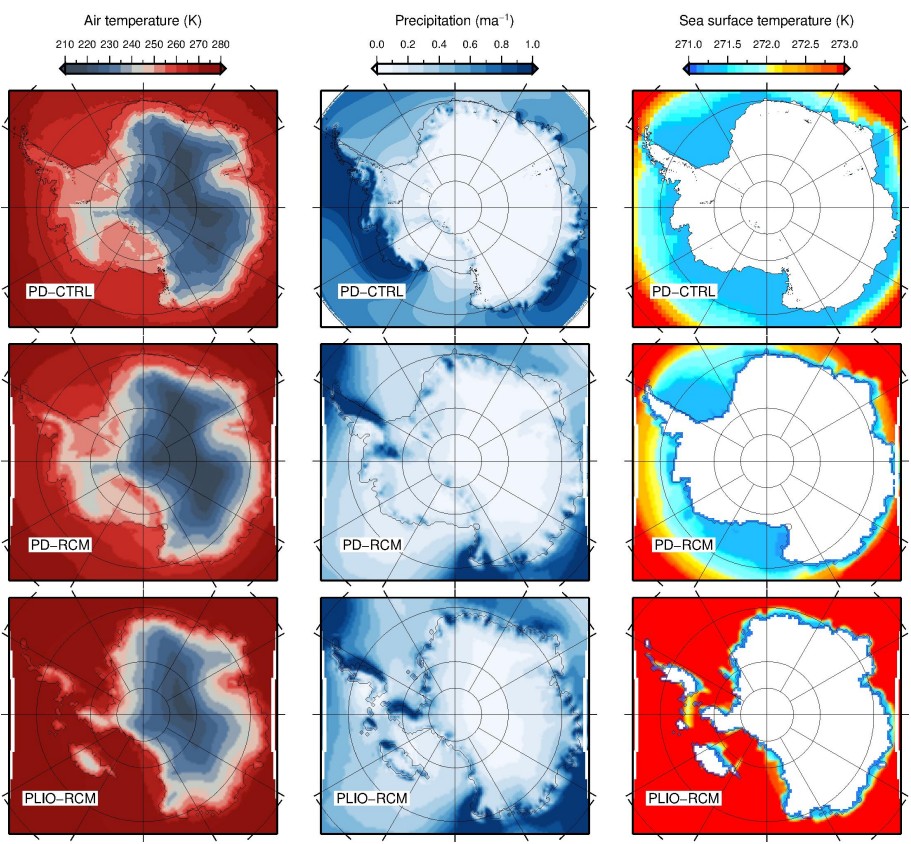

**Figure 2.** Air temperature, precipitation, and sea surface temperature used for our simulations. Top row: the 'control' (CTRL) scenario, employing gridded data from observations and modelling of present-day conditions (Comiso, 2000; Lenaerts et al., 2012; Thompson and Pollard, 1997; Pal et al., 2007). Middle row: present-day conditions as simulated by the Regional Climate Model. Bottom row: Pliocene conditions at 4.23 Ma as simulated by the Regional Climate Model.

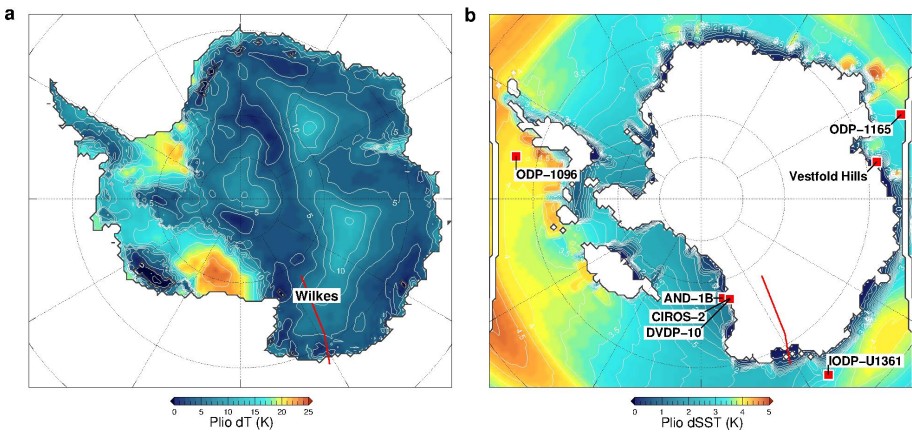

**Figure 3. a,** Air temperature anomaly for 4.23 Ma from regional climate modelling, adjusted for surface elevation differences between the RCM input orography and the present-day elevations used to initialize the ice-sheet model. Contours show 2.5° C increments. Biases used in certain simulations are additional to the warming values shown here. **b,** As **a** but showing sea-surface temperature anomaly. Contours show 0.25° C increments. Locations of profile shown in Fig. 7 also shown (red line), as well as the locations (red squares) of the palaeo-environmental proxy data in Table 1.

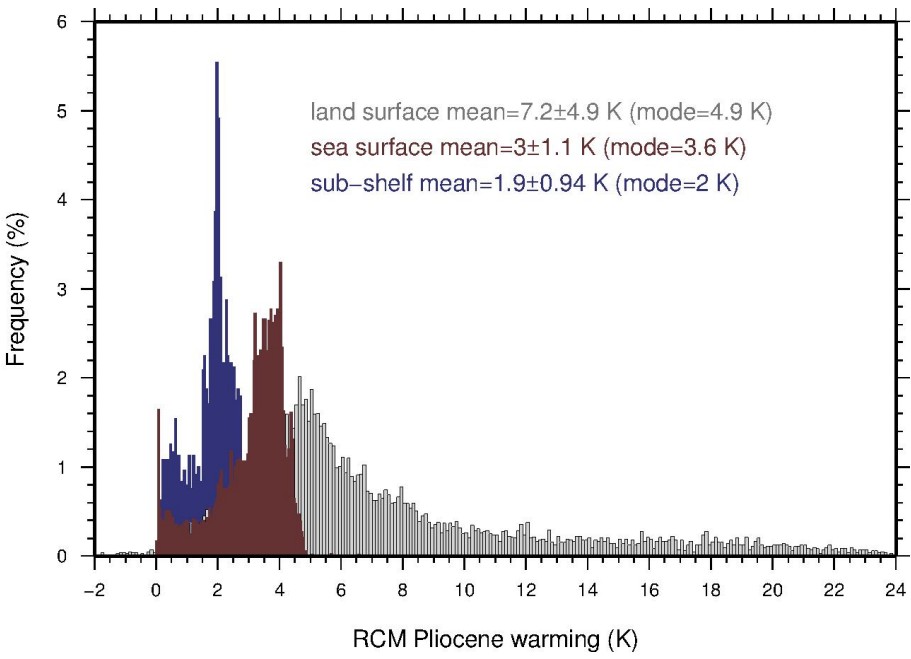

**Figure 4.** Histogram representation on the anomaly data shown in Fig. 3. Biases used in certain simulations are additional to the warming values shown here. Note the very long tail that skews the mean value of modelled land surface warming, indicating that the modal value might be most representative of the continent as a whole. Sea surface temperature anomalies are also higher in the open ocean than those of a subset representing ocean areas where present-day ice shelves exist.

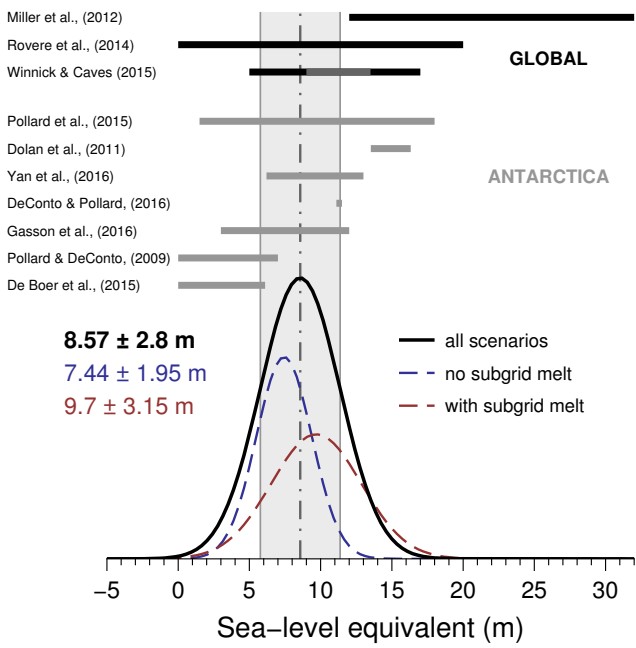

**Figure 5.** Simulated Antarctic ice sheet contributions to global mean sea level under a range of RCM-based climatologies, using two different grounding-line parameterisations. Values from previous studies are also shown (grey bars), as well as global sea-level highstand values inferred from far-field sites.

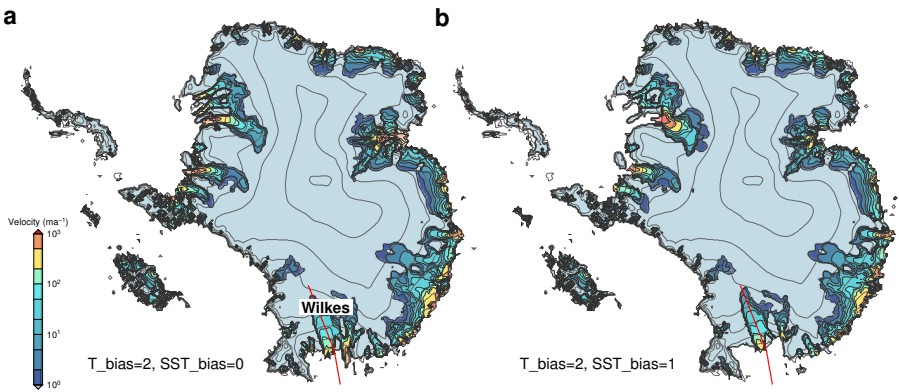

**Figure 6.** Basal ice velocities for the simulated early Pliocene ice sheet under environmental conditions from regional climate model simulations using two different bias adjustments. Fastest-flowing outlets occur in the subglacial basins and troughs of East Antarctica. These zones correspond to inferred areas of subglacial erosion during past warm climates (Cook et al., 2013; Aitken et al., 2016). Red line identifies catchment transect shown in Fig. 7.

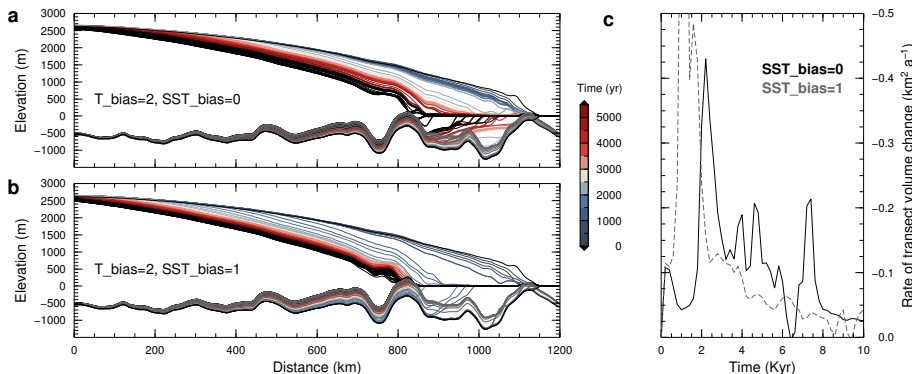

**Figure 7.** Mechanism and timescale of ice sheet retreat across Wilkes Subglacial Basin. **a,** Ice-sheet margin initially occupies stable location pinned on bedrock high, unaffected by marine influence. Gradual surface lowering from rising air temperatures promotes thinning, flotation, and subsequent rapid inland retreat of margin. **b,** Identical simulation to **a,** but employing a warmer ocean. Coloured lines denote ice geometries at 200-year intervals. Note also the associated bedrock uplift following ice retreat. **c,** Rate of retreat across the WSB is non-uniform, despite time-invariant forcings, and is governed primarily by the location of bedrock pinning points.

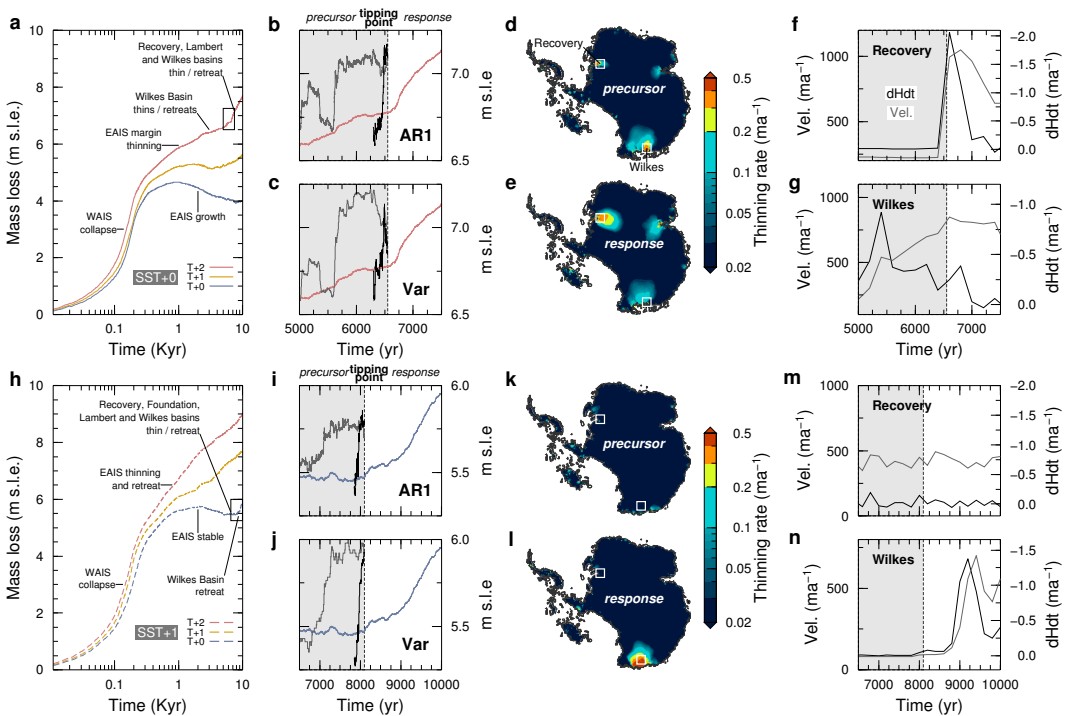

**Figure 8. a,** Sea-level equivalent ice mass loss for ice-sheet simulations forced by regional climate model outputs allowing for 0–2°C air temperature bias. Inset box identifies region shown in **b, c**. Precursor signatures of tipping points: **b,** increasing autocorrelation (AR1), and **c,** increasing variance (Var). Grey lines show precursors over 1500-year sliding window, black lines show precursors during the 250-year sliding window immediately preceding the tipping point. Additional detail shown in Figure 9. Mean annual thinning rate over precursor **d,** and response **e,** periods. **f, g,** Thinning rate (dHdt) and surface velocity (Vel.) at locations in the Recovery and Wilkes subglacial basins (white boxes in **d, e**). Note the lagged response of peak ice velocity compared to thinning maxima. Panels **h–n** as **a–g** but for scenarios with 1°C sea-surface temperature bias.

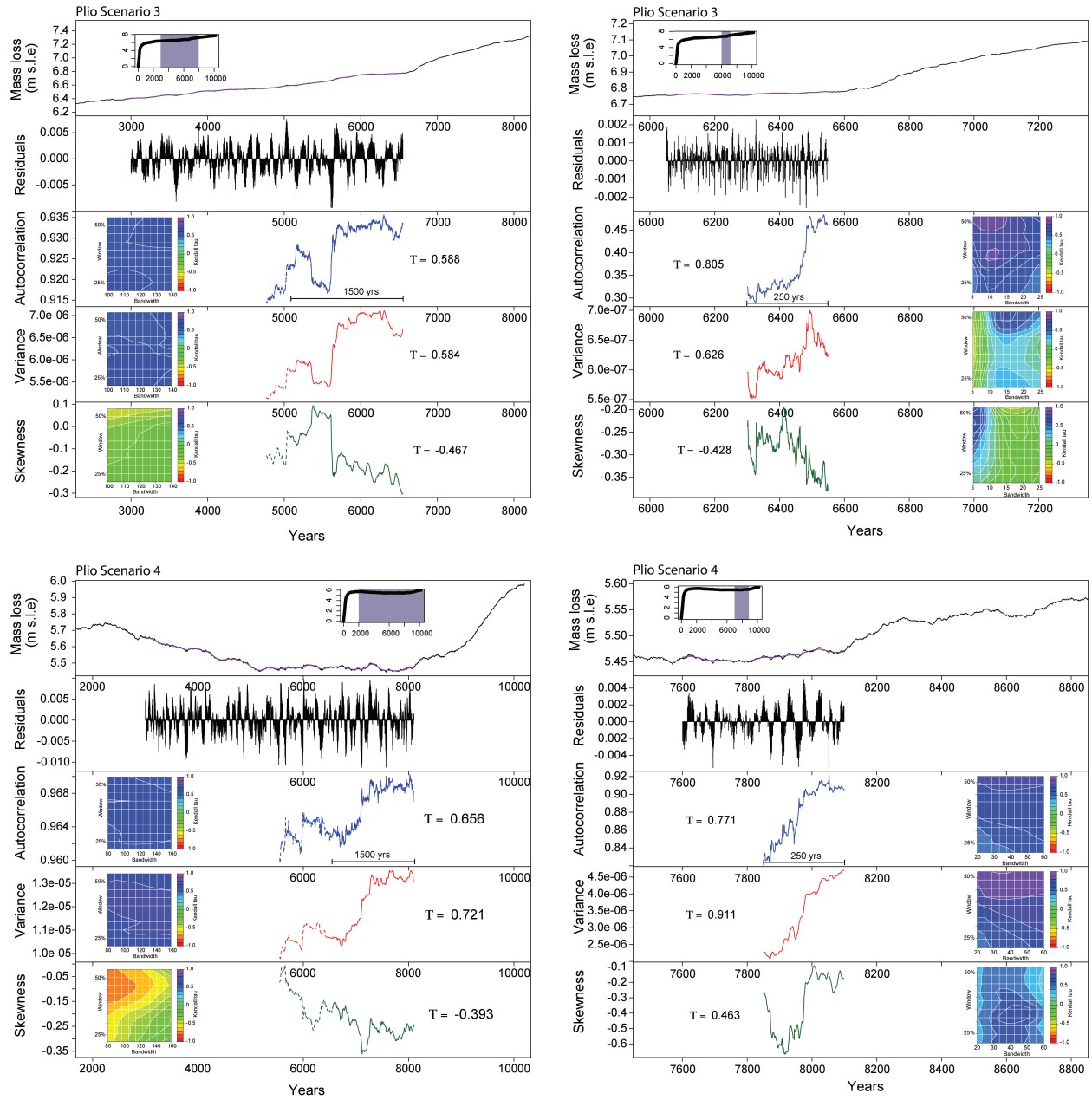

**Figure 9. | Tipping point analyses for two scenarios.** Upper row shows results of analyses of dT +2°C, dSST +0°C, lower row shows dT +0°C, dSST +1°C. In each panel, top row shows modelled mass loss curve with inset identifying period used in the tipping point analysis. The four lower rows each show one key indicator metric (residuals, autocorrelation, variance, skewness) as described in 'Methods'. Left and right columns show results based on 1500-year (from the solid line; dashed line shows analysis prior to this window) and 250-year sliding windows respectively. Note the steeper trends and higher Tau values in the shorter analyses, indicative of a stronger signal over the shorter timeframe. Inset contour plots display the range of Kendall tau values for different sliding window lengths (25-50%) and smoothing bandwidths (5-15%).

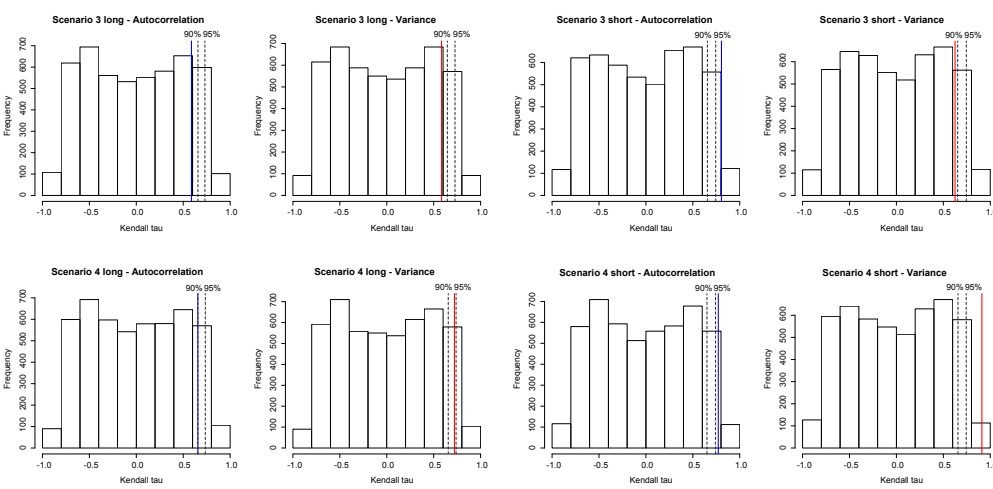

**Figure 10.** Histograms showing Kendall tau correlation coefficients for surrogate time series, generated from the two scenarios for which the existence of tipping points was investigated. Values relate to autocorrelation (blue lines) and variance (red lines) scores shown in Table 3. Dashed lines mark 90% and 95% confidence intervals.

## Appendix A

### A1

*Author contributions.* NRG devised and carried out the ice-sheet modelling experiments, and led the research. ZAT undertook the tipping point analysis. RHL, TRN, and RMM compiled and interpreted paleoenvironmental proxy data. EGWG and DEK undertook climate modelling experiments. All authors contributed to the development of ideas and writing of the manuscript.

*Competing interests.* The authors declare no conflict of interests.

*Acknowledgements.* We are grateful to two anonymous reviewers for their comments on a previous version of this manuscript. This work was funded by contract VUW1203 of the Royal Society of New Zealand's Marsden Fund, with support from the Antarctic Research Centre, Victoria University of Wellington, ANDRILL, GNS Science, NSF Polar Programs grants ANT-1043712 and PLR-1245899, and the Australian Research Council (ARC) including a Laureate Fellowship (FL100100195) supporting ZAT. Development of PISM is supported by NASA grants NNX13AM16G and NNX13AK27G.

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
