# Peer review of "Antarctic climate and ice sheet configuration during a peak-warmth Early Pliocene interglacial"

_Climate of the Past, 2016_

## Referee Comment (RC1) · Anonymous Referee #1 · 2 Jan 2017

The fate of the AIS in the future is of great importance owing to its capability to rise global sea-level by ∼60 m. Lack of long-term instrumental records hamper our understanding of the behavior of AIS, especially the EAIS, in the 21st century. Geological evidence and simulations for a past warmer-than-present world could advance our knowledge on how AIS may respond to a warmer climate. Golledge et al. investigated the AIS in the Pliocene that is frequently argued as a potential analogue for future world. Although numerous modeling works have been performed targeting at the Pliocene AIS, ranging from offline to fully coupled climate-ice sheet simulations, their work differs with previous ones mainly in the so-called "tipping point" analysis.

However, I have large concern on effectiveness and implication of the "tipping point"

[Figure]

analysis performed in this work. In my opinion, the level of warming needed to melt an ice sheet completely or a key region (e.g., ice over Wilkes Subglacial Basin in the Pliocene) is considered to be a critical threshold, or tipping point. For example, the tipping point for the Greenland ice sheet is about 1.6 oC (Robinson et al., 2012). The authors performed the so-called "tipping point" analysis, but give no efficient information on the actual tipping point. In addition, the technique used may be inapplicable here as the climatic forcing is constant. In this way, I think the signal detected is the time needed to melt parts of ice sheet for a given forcing, such as these shown in Fig. 8. Besides, as Wilkes Basin is a key region for the stability of the Pliocene AIS, it is necessary to analyze temporal evolution of ice volume over there and perform the "tipping point" analysis. [Robinson A, Calov R, Ganopolski A. Multistability and critical thresholds of the Greenland ice sheet. Nature Climate Change, 2012, 2(6): 429-432]

Other concern is on the uncertainty in the modeled AIS. The values of ice sheet model parameters are poorly constrained due to the limited observations over Antarctica, which may introduce an uncertainty into the simulated AIS. For example, Yan et al. (2016) indicated that the largest source of uncertainty in the modeled Pliocene AIS is derived from ice sheet model parameters, which result in a range of 10.8 m in sea level equivalent. I recommend that the authors should perform several sensitivity runs to test whether the so-called "tipping point" is greatly affected by parameter uncertainty. [Yan, Q., Z. Zhang, and H. Wang (2016), Investigating uncertainty in the simulation of the Antarctic ice sheet during the mid-Piacenzian, J. Geophys. Res. Atmos., 121, 1559–1574, doi:10.1002/2015JD023900].

Additionally, the simulated absolute temperatures with RCM are generally consistent with proxies, though a bias of 1∼2 oC is found. So I think it is useful to drive the PISM with outputs from the RCM directly. However, the authors employ an "anomaly" method to construct the Pliocene forcing used in PISM. The method should be justified. The authors can also compare the simulated temperature anomaly with reconstructed anomaly or compare the newly constructed Pliocene forcing with reconstructions. In

this way, they can test which method is better, the "direct method" or the "anomaly method".

Specific comments:

Page 2, line 4: it should be "2-4 oC" warmer in the mid-Pliocene.

Page 2, line 23: How the sea surface temperature is set over land? It is set to land temperature or others? The temperature over subglacial basins are important and affect the simulated ice sheet retreat.

Page 4, line 25: please add a brief description on the parameterizations of sub-shelf melting in PISM.

Page 5, line 24: how long the model is integrated? 10 kyr? Does the model reach quasi-equilibrium? Please clarify this in the manuscript.

In Fig. 4: How the temperature anomaly over sub-shelf region is calculated? Is WAIS also removed in the control run? Actually, the RCM used cannot simulate oceanic temperature below ice shelves that is required in PISM.

In Fig. 5: How many experiments are carried out? Nine? If so, as the number of experiment is not large enough, the results from each experiment can be plotted as a dot rather than dashed lines in Fig. 5, which may cause misunderstanding. Besides, the work of Yan et al. (2016, JGR) can be added here.

---

## Referee Comment (RC2) · Anonymous Referee #2 · 3 Jan 2017

The behaviors of different parts of the Antarctica ice sheet were analyzed by using an ice sheet model which was forced by climate fields based on a regional model outputs. Simulations were performed under several scenarios of surface air temperature and sea surface temperature either directly from the regional model outputs or by adding 1 or 2 degree C according to proxy reconstructions. A major focus is on the tipping point analysis of the Antarctica ice sheet evolution under a constant climate forcing. I find the sensitivity analyses in this paper interesting and helpful for understanding the Antactica ice sheet dynamics in a warmer-than-present climate.

As in many early Pliocene climate simulations, a constant insolation forcing was unfortunately used in the climate simulation of this paper although the warm periods of the

Pliocene cover many precession and obliquity cycles. The insolation at 4.23Ma was used because the austral summer insolation reaches a maximum. However, this kind of insolation does not necessarily lead to a warmer condition globally or even over the Antarctica region. For example, it was shown in Yin and Berger (Individual contribution of insolation and CO2 to the interglacial, Clim Dyn, 2012, 38:709–724) that, probably due to their much higher BOREAL summer insolation, the interglacials MIS-5e and MIS-15 had a warmer Antarctica than the reference experiment although they had a lower austral summer insolation. Using one constant insolation forcing could induce uncertainty in the climate simulation. It might be one of the reasons that the simulated temperature is cooler than the reconstructed one. However, as the focus of this paper is on the tipping point analysis under a constant forcing, using the insolation forcing only at 4.23Ma would be acceptable, but the authors need to point out the limitation of their forcing selection especially when they also try to estimate the absolute contribution of Antarctica ice sheet to sea level.

The authors have run the ice sheet model for 10,000 years with a constant climate forcing. Any reason for choosing such a long time period? The inflections (tipping points) on Fig8 occur very late in the simulations. We wouldn't see them if the simulation length were not long enough. It seems unreasonable to run the ice sheet model for 10,000 years with a constant forcing because insolation reaches another extreme in 10,000 years. Moreover, under such a constant and warm condition, the ice sheet would never reach equilibrium, as confirmed by the mass loss curves in Fig8.

My impression is that the focus of this paper is on the tipping point analysis of the Antarctica ice sheet evolution under several climate scenarios warmer than present, so the title of this paper seems not precise.

The ice sheet model is not interactively coupled with climate model. The potential influence on the results should be mentioned.

In which degree would the results be affected by the initial ice sheet condition?

More information should be given on the GENESIS model, on the regional model and on experiment design.

Is the subsurface ocean temperature considered in this ice sheet simulation and how?

The role of precipitation is not much mentioned in the paper. Is it because precipitation is not important?

Page 2, line 28: temperature anomalies as inputs or temperature anomalies plus present-day observation?

Page 2, line 30: in the climate model, the WAIS is already removed. Is a circular reasoning involved here?

Page 7, line 24: please explain what the climate-topography thresholds mean.

Page 8, line 14: It seems that there is one tipping point on the yellow line in Fig8a.

Figure 1: The reference of the astronomical data should be cited.

---

## Author Comment (AC2) · 20 Feb 2017

The behaviors of different parts of the Antarctica ice sheet were analyzed by using an ice sheet model which was forced by climate fields based on a regional model outputs. Simulations were performed under several scenarios of surface air temperature and sea surface temperature either directly from the regional model outputs or by adding 1 or 2 degree C according to proxy reconstructions. A major focus is on the tipping point analysis of the Antarctica ice sheet evolution under a constant climate forcing.

>>> It was not our intention that the tipping point analysis should be seen as a 'major focus', in fact we included it as an interesting additional tool that could shed light on long-term behaviour of the ice sheet, but no more. In the revision we can make this

clearer.

I find the sensitivity analyses in this paper interesting and helpful for understanding the Antactica ice sheet dynamics in a warmer-than-present climate.

>>> We are grateful to the reviewer for this comment and glad that these elements are useful.

As in many early Pliocene climate simulations, a constant insolation forcing was unfortunately used in the climate simulation of this paper although the warm periods of the Pliocene cover many precession and obliquity cycles. The insolation at 4.23Ma was used because the austral summer insolation reaches a maximum. However, this kind of insolation does not necessarily lead to a warmer condition globally or even over the Antarctica region. For example, it was shown in Yin and Berger (Individual contribution of insolation and CO2 to the interglacial, Clim Dyn, 2012, 38:709–724) that, probably due to their much higher BOREAL summer insolation, the interglacials MIS-5e and MIS-15 had a warmer Antarctica than the reference experiment although they had a lower austral summer insolation. Using one constant insolation forcing could induce uncertainty in the climate simulation. It might be one of the reasons that the simulated temperature is cooler than the reconstructed one. However, as the focus of this paper is on the tipping point analysis under a constant forcing, using the insolation forcing only at 4.23Ma would be acceptable, but the authors need to point out the limitation of their forcing selection especially when they also try to estimate the absolute contribution of Antarctica ice sheet to sea level.

>>> We are very grateful to the reviewer for bringing this to our attention, and indeed this does perhaps offer an additional explanation for why our RCM underestimates apparent temperatures. In the revision we will expand this aspect of the text to acknowledge this, and will include the reference to Yin and Berger, 2012. We will also clarify more explicitly that changes in insolation forcing through the interglacial mean that our sea-level estimates after 10kyr are upper bounds, with respect to the applied

forcing.

The authors have run the ice sheet model for 10,000 years with a constant climate forcing. Any reason for choosing such a long time period? The inflections (tipping points) on Fig8 occur very late in the simulations. We wouldn't see them if the simulation length were not long enough. It seems unreasonable to run the ice sheet model for 10,000 years with a constant forcing because insolation reaches another extreme in 10,000 years. Moreover, under such a constant and warm condition, the ice sheet would never reach equilibrium, as confirmed by the mass loss curves in Fig8.

>>> We acknowledge that the length of the simulations is beyond what is likely for an interglacial peak in insolation, but we defined the period in order to capture a 'typical' interglacial, which are usually of this order. The reasons for doing so, and for maintaining a constant forcing for this period, was to elicit the long-term response of the ice sheet and to investigate whether any non-linear behaviour may occur even under constant forcing. We found that under certain scenarios such non-linearities do in fact take place. Whilst the model set-up may seem unrealistic, the experimental results nonetheless shed light on how the AIS may behave under prolonged warming, which may be relevant to future scenarios, particularly if CO2 emissions act to delay the onset of the next glacial cooling (e.g. Ganopolski et al., 2016).

My impression is that the focus of this paper is on the tipping point analysis of the Antarctica ice sheet evolution under several climate scenarios warmer than present, so the title of this paper seems not precise.

>>> Actually the bulk of the new work (RCM simulations and ice sheet modelling) is focused on establishing the likely Antarctic climate during this particular interglacial, for which we also bring in substantial proxy evidence, and also the likely response of the ice sheet to this forcing. The tipping point analysis is somewhat secondary, as explained above. We therefore feel that the title is correct as it currently stands.

The ice sheet model is not interactively coupled with climate model. The potential

influence on the results should be mentioned.

>>> We will acknowledge this limitation more explicitly in the revised manuscript.

In which degree would the results be affected by the initial ice sheet condition?

>>> This is something that would be hard to quantify without re-running the entire ensemble, but we will certainly add some text to the revised manuscript to highlight this aspect.

More information should be given on the GENESIS model, on the regional model and on experiment design.

>>> OK, we will add further information on these aspects.

Is the subsurface ocean temperature considered in this ice sheet simulation and how?

>>> Our ice sheet model can only read in ocean temperature fields in the $x, y$ plane, so we do not account for temperature variations with depth. However, the basal melt scheme that we employ *does* account for the pressure effects of depth, leading to higher melt rates at deeper grounding lines than beneath the outer parts of the ice shelves. We will acknowledge this simplification in our approach in the revised manuscript.

The role of precipitation is not much mentioned in the paper. Is it because precipitation is not important?

>>> The precipitation anomalies are shown in Fig. 2, and their effects in terms of how they offset (or not) losses arising from warmer temperatures are described on pages 7 (line 34) and page 8 (line 2). We will add further information on the temperature / precipitation relationship in the revised manuscript and make the relative effects clearer.

Page 2, line 28: temperature anomalies as inputs or temperature anomalies plus present-day observation?

>>> Anomalies are added to the present-day field. We will clarify this in the text.

Page 2, line 30: in the climate model, the WAIS is already removed. Is a circular reasoning involved here?

>>> Yes possibly, and we dedicate four lines on this page and the next to acknowledging that this is an imperfect set up. Our approach tries to 'reverse' the air temperature increase arising from the removal of WAIS in the RCM by lapsing the temperature back to the present-day ice elevation, but we cannot do much about the ocean temperatures. The only alternative would be to start with a present-day geometry and employ a 2-way coupling between the ice sheet and the climate model to account for climate changes as the ice sheet evolves. This kind of work is at the forefront of this field of science and is being addressed by modelling groups around the world, but is non-trivial. At present we can only achieve a 1-way coupling and so are honest about the limitations that this comes with.

Page 7, line 24: please explain what the climate-topography thresholds mean.

>>> OK, we will better explain this in the revised manuscript.

Page 8, line 14: It seems that there is one tipping point on the yellow line in Fig8a.

>>> We undertook the tipping point analyses on all 6 timeseries shown in Fig. 8a,h. However, although visually it looks that there may be a tipping point in the yellow curve, statistically this does not have the signatures of critical slowing down such as an increase in autocorrelation and standard deviation. Consequently we chose to focus on other examples where the signatures were more robust. Our study did not set out to 'prove' that tipping points exist, rather it tried to simply establish if evidence for them could be found, and if so, under what conditions these behaviours may arise. Our manuscript text therefore describes all the scenarios, as well as the basis for our focusing on the two clearest examples.

Figure 1: The reference of the astronomical data should be cited.

>>> Yes, we were remiss not to include this and will correct this oversight immediately. The data comes from Laskar et al., 2004.

---

## Author Response (AR1)

The fate of the AIS in the future is of great importance owing to its capability to rise global sea-level by 60 m. Lack of long-term instrumental records hamper our understanding of the behavior of AIS, especially the EAIS, in the 21st century. Geological evidence and simulations for a past warmer-than-present world could advance our knowledge on how AIS may respond to a warmer climate. Golledge et al. investigated the AIS in the Pliocene that is frequently argued as a potential analogue for future world. Although numerous modeling works have been performed targeting at the Pliocene AIS, ranging from offline to fully coupled climate-ice sheet simulations, their work differs with previous ones mainly in the so-called "tipping point" analysis.

>>> We agree that our work is novel partly because of the tipping point analysis, but this is actually only a small part of the study, not the main focus. In beginning this work, our main point of difference compared to other studies was that we decided to focus on the early Pliocene, rather than the mid Pliocene, because the early Pliocene is very rarely (if at all) studied, and yet has interglacials characterised by summer insolation that is greater than occurs later in the Pliocene. In the revision we have improved the clarity of our introductory text so that this is clear.

However, I have large concern on effectiveness and implication of the "tipping point" analysis performed in this work. In my opinion, the level of warming needed to melt an ice sheet completely or a key region (e.g., ice over Wilkes Subglacial Basin in the Pliocene) is considered to be a critical threshold, or tipping point. For example, the tipping point for the Greenland ice sheet is about 1.6 oC (Robinson et al., 2012). The authors performed the so-called "tipping point" analysis, but give no efficient information on the actual tipping point.

>>> We would like to point out a fundamental difference between a tipping point and a threshold, terms that the reviewer uses synonymously. A tipping point is a transient feature, whereas a threshold is non-temporal. During the evolution of an ice sheet it may be that a tipping point is reached in which the trajectory of evolution changes. This is what our study investigates. This is not the same as defining a single temperature at which the ice sheet may be stable or unstable in a given area. Consequently it is not possible from our tipping point analysis to provide information on a threshold, the two phenomenon are simple different entities. In our revised manuscript we have added new text to make this distinction clearer, and refer the reader to a recent paper that *does* quantify threshold temperatures for individual catchments (Golledge et al., 2017, GRL).

In addition, the technique used may be inapplicable here as the climatic forcing is constant.

>>> Actually, this is the whole point of the tipping point analysis, and is what makes it so useful. If we had imposed a time-varying forcing, then accelerations in the mass loss of our simulated ice sheets could be attributable to changes in the forcing. This is *not* what we show. By analysing the timeseries data from a constant forcing experiment, the tipping point analysis is able to not only show where genuine system instabilities occur, but also to provide information on the timescale over which this instability evolves. A detailed explanation of tipping point analysis can be found in Thomas, 2016, QSR. In our revision we have clarified these aspects so that the utility of our approach is evident.

In this way, I think the signal detected is the time needed to melt parts of ice sheet for a given forcing, such as these shown in Fig. 8.

>>> Yes indeed, the analysis shows that gradual surface lowering leads to a critical point at which margin destabilisation takes place, and rapid mass loss ensues.

Besides, as Wilkes Basin is a key region for the stability of the Pliocene AIS, it is necessary to analyze temporal evolution of ice volume over there and perform the "tipping point" analysis. [Robinson A, Calov R, Ganopolski A. Multistability and critical thresholds of the Greenland ice sheet. Nature Climate Change, 2012, 2(6): 429-432]

>>> The reviewer is correct to point out that the Wilkes basin is a key area, and for this reason we dedicated text and figures to the specific investigation of ice dynamics in this area. We show clearly that initial surface lowering from atmospheric warming leads eventually to thinning and flotation, and subsequent grounding-line retreat that is accompanied by mass loss. However, we do not apply the tipping point analysis to this sector in isolation, because 1) we have already shown clearly how this area responds to forcing, and 2) our purpose with the tipping point analysis is to see how these various regional sensitivities play out in terms of a continentally-integrated value (which is ultimately what is important for sea level rise).

Other concern is on the uncertainty in the modeled AIS. The values of ice sheet model parameters are poorly constrained due to the limited observations over Antarctica, which may introduce an uncertainty into the simulated AIS. For example, Yan et al. (2016) indicated that the largest source of uncertainty in the modeled Pliocene AIS is derived from ice sheet model parameters, which result in a range of 10.8 m in sea level equivalent. I recommend that the authors should perform several sensitivity runs to test whether the so-called "tipping point" is greatly affected by parameter uncertainty. [Yan, Q., Z. Zhang, and H. Wang (2016), Investigating uncertainty in the simulation of the Antarctic ice sheet during the mid-Piacenzian, J. Geophys. Res. Atmos., 121, 15591574, doi:10.1002/2015JD023900].

>>> There are two approaches to dealing with ice model parameter uncertainty in these kind of studies. One approach simply undertakes thousands of experiments (a large ensemble) with incremental changes in each of several key parameters, such as flow enhancement factors. The results are then subsequently analysed with respect to observational constraints to establish which ensemble members are consistent with the data. We do *not* adopt this kind of approach. Instead we follow a more targeted methodology in which model parameter choice is incrementally refined through an iterative procedure in which we constrain our model to fit the present-day ice sheet geometry and surface velocity field. To achieve a good fit we adjust ice flow parameters based on expert judgement, not in an unguided manner as is done with ensemble approaches. The result is a spun-up, thermally and dynamically equilibrated ice sheet simulation that is the best fit to observational constraints that is possible by tuning available model parameters. All our Pliocene experiments are run from this starting point. Thus whilst we agree with the reviewer that parameter uncertainty can be a large source of error, we argue that our approach removes or significantly reduces this uncertainty prior to our undertaking the prognostic experimentation. In our revision we have added new text to clarify this, and have reiterated our tuning procedure which we have described previously in other papers (e.g Golledge et al., 2015, Nature).

Additionally, the simulated absolute temperatures with RCM are generally consistent with proxies, though a bias of 1-2 oC is found. So I think it is useful to drive the PISM with outputs from the RCM directly. However, the authors employ an "anomaly" method to construct the Pliocene forcing used in PISM. The method should be justified. The authors can also compare the simulated temperature anomaly with reconstructed anomaly or compare the newly constructed Pliocene forcing with reconstructions. In this way, they can test which method is better, the "direct method" or the "anomaly method".

>>> We use the 'raw' RCM temperatures but have to implement them as anomalies to the present-day field because the model is tuned with respect to the latter. Using the RCM temperatures directly without correcting for WAIS loss or the bias of the RCM with respect to the present-day climate would perhaps yield an interesting simulation, but it would have no value

for our study. To use the RCM simulation in it's raw state we would need to also run our tuning procedure and full spinup with the RCM temperature field, but since we know that this field is not correct (i.e. it isn't identical to observed data) this approach seems distinctly less preferable than the anomaly approach we employ. We also use the RCM values with additional biases added, to account for uncertainties as described in the paper. But the simulations using the un-adjusted RCM fields are there explicitly for comparison, so the reader can assess for themselves what the impacts of the adjustments are. We have made sure that both of these aspects are now even clearer in our revised manuscript.

Specific comments: Page 2, line 4: it should be "2-4 oC" warmer in the mid-Pliocene.

>>> OK.

Page 2, line 23: How the sea surface temperature is set over land? It is set to land temperature or others? The temperature over subglacial basins are important and affect the simulated ice sheet retreat.

>>> We apologies for not making this clear. For areas that are ice-covered at the start of the run we prescribe a uniform temperature of 271.2 K, essentially the sea-water freezing point. This avoids potential errors that could be introduced by interpolating ocean fields landward, but we recognise that this may lead to underestimated basal melt in subglacial basins during ice sheet retreat. In the revision we have added new text to make this clearer.

Page 4, line 25: please add a brief description on the parameterizations of sub-shelf melting in PISM.

>>> OK, we have added new text describing this.

Page 5, line 24: how long the model is integrated? 10 kyr? Does the model reach quasi-equilibrium? Please clarify this in the manuscript.

>>> Page 5, line 29-30: "Simulations are run for 10 kyr." Figure 8 illustrates the evolution of the ice sheet in the range of scenarios discussed. From this figure it can be seen that under some climatologies a near-equilibrium state is reached, whilst under others it is not. This is described in the Results section, on pages 7 & 8. We have now also made it explicitly clear that almost all simulations approach a steady-state within the 10 kyr run.

In Fig. 4: How the temperature anomaly over sub-shelf region is calculated? Is WAIS also removed in the control run? Actually, the RCM used cannot simulate oceanic temperature below ice shelves that is required in PISM.

>>> The ocean forcing applied to the ice sheet model from the RCM simulations is shown in Figure 2 and described in the text. The RCM simulation uses a geometry that does not include WAIS. This does not affect the grounded ice, however, since the temperature field is only used to compute basal shelf melt. In the simulations that use the sub-grid basal melt interpolation, this temperature field will also result in melt at the grounding line (see page 5 for explanation).

In Fig. 5: How many experiments are carried out? Nine? If so, as the number of experiment is not large enough, the results from each experiment can be plotted as a dot rather than dashed lines in Fig. 5, which may cause misunderstanding. Besides, the work of Yan et al. (2016, JGR) can be added here.

>>> In total we present results from 18 experiments, in which we separate out the two suites that use or do not use the sub-grid melt scheme. The solid black line represents all members of the ensemble, whilst the thinner lines represent the two components. These latter curves are provided simply to illustrate their relationship to the ensemble curve, and we do not ascribe any statistical significance to them particularly, they simply offer an additional level of detail that

helps the reader identify the consequences of the two grounding-line schemes. We have added the values from Yan et al. 2016.

The behaviors of different parts of the Antarctica ice sheet were analyzed by using an ice sheet model which was forced by climate fields based on a regional model outputs. Simulations were performed under several scenarios of surface air temperature and sea surface temperature either directly from the regional model outputs or by adding 1 or 2 degree C according to proxy reconstructions. A major focus is on the tipping point analysis of the Antarctica ice sheet evolution under a constant climate forcing.

>>> It was not our intention that the tipping point analysis should be seen as a 'major focus', in fact we included it as an interesting additional tool that could shed light on long-term behaviour of the ice sheet, but no more. In the revision we have rewritten parts of the introductory text to make this clearer.

I find the sensitivity analyses in this paper interesting and helpful for understanding the Antactica ice sheet dynamics in a warmer-than-present climate.

>>> We are grateful to the reviewer for this comment and glad that these elements are useful.

As in many early Pliocene climate simulations, a constant insolation forcing was unfortunately used in the climate simulation of this paper although the warm periods of the Pliocene cover many precession and obliquity cycles. The insolation at 4.23Ma was used because the austral summer insolation reaches a maximum. However, this kind of insolation does not necessarily lead to a warmer condition globally or even over the Antarctica region. For example, it was shown in Yin and Berger (Individual contribution of insolation and CO2 to the interglacial, Clim Dyn, 2012, 38:709724) that, probably due to their much higher BOREAL summer insolation, the interglacials MIS-5e and MIS-15 had a warmer Antarctica than the reference experiment although they had a lower austral summer insolation. Using one constant insolation forcing could induce uncertainty in the climate simulation. It might be one of the reasons that the simulated temperature is cooler than the reconstructed one. However, as the focus of this paper is on the tipping point analysis under a constant forcing, using the insolation forcing only at 4.23Ma would be acceptable, but the authors need to point out the limitation of their forcing selection especially when they also try to estimate the absolute contribution of Antarctica ice sheet to sea level.

>>> We are very grateful to the reviewer for bringing this to our attention, and indeed this does perhaps offer an additional explanation for why our RCM underestimates apparent temperatures. In the revision we have expanded this aspect of the text to acknowledge that $CO_2$ and insolation play time-varying roles in determining air temperature, and have included the reference to Yin and Berger, 2012. We have also clarified more explicitly that changes in insolation forcing through the interglacial mean that our sea-level estimates after 10kyr are upper bounds, with respect to the applied forcing.

The authors have run the ice sheet model for 10,000 years with a constant climate forcing. Any reason for choosing such a long time period? The inflections (tipping points) on Fig8 occur very late in the simulations. We wouldnt see them if the simulation length were not long enough. It seems unreasonable to run the ice sheet model for 10,000 years with a constant forcing because insolation reaches another extreme in 10,000 years. Moreover, under such a constant and warm condition, the ice sheet would never reach equilibrium, as confirmed by the mass loss curves in Fig8.

>>> We acknowledge that the length of the simulations is beyond what is likely for an interglacial peak in insolation, but we defined the period in order to capture a 'typical' interglacial, which are usually of this order. The reasons for doing so, and for maintaining a constant forcing for this period, was to elicit the long-term response of the ice sheet and to investigate whether

any non-linear behaviour may occur even under constant forcing. We found that under certain scenarios such non-linearities do in fact take place. Whilst the model set-up may seem unrealistic, the experimental results nonetheless shed light on how the AIS may behave under prolonged warming, which may be relevant to future scenarios, particularly if CO2 emissions act to delay the onset of the next glacial cooling (e.g. Ganopolski et al., 2016). We have now added text to the 'Introduction' to acknowledge that the 10 kyr run is not entirely realistic, and that we justify this on the basis of trying to learn something new about longer-term ice sheet behaviour.

My impression is that the focus of this paper is on the tipping point analysis of the Antarctica ice sheet evolution under several climate scenarios warmer than present, so the title of this paper seems not precise.

>>> Actually the bulk of the new work (RCM simulations, ice sheet modelling, compilatiotn of palaeo-environmental proxy data) is focused on establishing the likely Antarctic climate during this particular interglacial, and also the likely response of the ice sheet to this forcing. The tipping point analysis is somewhat secondary, as explained above. We therefore feel that the title was correct as it previously stood, however, in light of the previous comments regarding the possibility that peak insolation does not necessarily equate to peak warmth, we have modified the title to be more specific.

The ice sheet model is not interactively coupled with climate model. The potential influence on the results should be mentioned.

>>> We have acknowledged this limitation more explicitly in the revised manuscript.

In which degree would the results be affected by the initial ice sheet condition?

>>> This is something that would be hard to quantify without re-running the entire ensemble, but we have now added new text to the revised manuscript to highlight this aspect, and the inference that initial conditions would most likely affect rates of change in the early parts of each simulation, rather than the equilibrium response.

More information should be given on the GENESIS model, on the regional model and on experiment design.

>>> OK, we have now added further information on these aspects.

Is the subsurface ocean temperature considered in this ice sheet simulation and how?

>>> The RCM only outputs SST values, and our ice sheet model can only read in ocean temperature fields in the $x, y$ plane, so we do not account for temperature variations with depth. However, the basal melt scheme that we employ *does* account for the pressure effects of depth, leading to higher melt rates at deeper grounding lines than beneath the outer parts of the ice shelves. We have now acknowledged this simplification in our approach in the revised manuscript, and refer the reader to Golledge et al., 2017, GRL, in which a figure is presented that shows basal melt rates and how they vary with depth.

The role of precipitation is not much mentioned in the paper. Is it because precipitation is not important?

>>> The precipitation anomalies are shown in Fig. 2, and their effects in terms of how they offset (or not) losses arising from warmer temperatures are described on pages 7 (line 34) and page 8 (line 2). We have now added further information on the temperature / precipitation parameterisation in the revised manuscript.

Page 2, line 28: temperature anomalies as inputs or temperature anomalies plus present-day observation?

$>>>$ Anomalies are added to the present-day field. We have now clarified this in the text.

Page 2, line 30: in the climate model, the WAIS is already removed. Is a circular reasoning involved here?

$>>>$ Yes possibly, and we dedicated four lines on that page and the next to acknowledge that this is an imperfect set up. Our approach tries to 'reverse' the air temperature increase arising from the removal of WAIS in the RCM by lapsing the temperature back to the present-day ice elevation, but we cannot do much about the ocean temperatures. The only alternative would be to start with a present-day geometry and employ a 2-way coupling between the ice sheet and the climate model to account for climate changes as the ice sheet evolves. This kind of work is at the forefront of this field of science and is being addressed by modelling groups around the world, but is non-trivial. At present we can only achieve a 1-way coupling and so are honest about the limitations that this comes with.

Page 7, line 24: please explain what the climate-topography thresholds mean.

$>>>$ OK, we have rewritten this sentence to make this clearer in the revised manuscript.

Page 8, line 14: It seems that there is one tipping point on the yellow line in Fig8a.

$>>>$ We undertook the tipping point analyses on all 6 timeseries shown in Fig. 8a,h. However, although visually it looks that there may be a tipping point in the yellow curve, statistically this does not have the signatures of critical slowing down such as an increase in autocorrelation and standard deviation. Consequently we chose to focus on other examples where the signatures were more robust. Our study did not set out to 'prove' that tipping points exist, rather it tried to simply establish if evidence for them could be found, and if so, under what conditions these behaviours may arise. Our manuscript text therefore describes all the scenarios, as well as the basis for our focusing on the two clearest examples.

Figure 1: The reference of the astronomical data should be cited.

$>>>$ Yes, we were remiss not to include this and have corrected this oversight. The data comes from Laskar et al., 2004.

[revised manuscript text omitted]